# Description, Distribution, and Relevance of Viruses of the Forest Pathogen *Gremmeniella abietina*

**DOI:** 10.3390/v10110654

**Published:** 2018-11-20

**Authors:** Leticia Botella, Jarkko Hantula

**Affiliations:** 1Phytophthora Research Centre, Department of Forest Protection and Wildlife Management, Faculty of Forestry and Wood Technology, Mendel University in Brno, Zemědělská 1, 613 00 Brno, Czech Republic; 2Forest Health and Biodiversity, Natural Resources Institute Finland (Luke), Latokartanonkaari 9, 00790 Helsinki, Finland; jarkko.hantula@luke.fi

**Keywords:** *Brunchorstia pinea*, conifers, mycovirus, dsRNA, ssRNA, phylogeny, evolution

## Abstract

The European race of the ascomycetous species *Gremmeniella abietina* (Lagerberg) Morelet includes causal agents of shoot blight and stem canker of several conifers in Europe and North America, which are known to host a diverse virome. GaRV6 is the latest and sixth mycovirus species reported within *G. abietina*. Before its description, one victorivirus and one gammapartitivirus species were described in biotype A, two mitoviruses in both biotypes A and B and a betaendornavirus in biotype B. Possible phenotypic changes produced by mycoviruses on *G. abietina* mycelial growth have been reported in Spanish mitovirus-free and GaRV6-hosting *G. abietina* isolates, which had higher growth rates at the optimal temperature of 15 °C, but no other major differences have been observed between partitivirus-like dsRNA and dsRNA-free isolates. In this review, we reappraise the diversity of viruses found in *G. abietina* so far, and their relevance in clarifying the taxonomy of *G. abietina*. We also provide evidence for the presence of two new viruses belonging to the families *Fusariviridae* and *Endornaviridae* in Spanish isolates.

## 1. Taxonomy of *G. abietina* and Relevance in Forestry

*Gremmeniella abietina* is a virulent haploid ascomycete responsible for shoot dieback and Scleroderris canker on conifers including spruces, firs, larches, pines, and junipers in North, Central, and South Europe, northeastern North America, and East Asia [1,2,3,4,5,6].

The taxonomy of *G. abietina* and its relation to forestry is a complex issue, since the taxon is divided into a number of varieties, races, and biotypes. These include two varieties: *G. abietina* var. *abietina* that mainly affects pines, and *G. abietina* var. *balsamea* that attacks firs and spruces [7]. Within *G. abietina* var. *abietina*, three races—Asian, North American, and European—were described based on serological analyses [8]. It has also been proposed that these races would be considered as separate species [9,10]. In the European race, three biotypes have been identified on the basis of symptoms, septa numbers, spore length, and molecular markers; namely, the alpine biotype, biotype A (LTT, large tree type), and biotype B (STT, small tree type) [11,12,13,14]. Furthermore, there is a distinctive population of *G. abietina* in Spain that may stem from biotype A of the European race [15,16].

In a taxonomic context, obligate parasites without an extracellular phase, such as mycoviruses, may be considered especially informative because they can only spread through mycelial contacts [17]. Therefore, determining the presence and evolutionary history of mycoviruses in different populations of *G. abietina* will shed light on the origin and pathways of its spread. In the literature, there are a number of examples that support the use of mycoviruses as tracers of the origin and pathways of different plant pathogens, i.e., *Hymenoscyphus fraxineus* [18], *Cryphonectria parasitica* [19] and *Heterobasidion annosum* [20].

The study of mycoviruses may also clarify the level of fungal compatibility between species. There is growing evidence suggesting differences in the fungal incompatibility rate [21], and a number of cases of mycovirus co-specificity in phylogenetically separated fungi have been reported, not only in the laboratory, but also in nature [20,22,23,24,25,26,27,28,29].

## 2. Occurrence of Viruses in *G. abietina*: Description of Their Genome and Structure, and Phylogenetic Relationships

The presence of putative mycoviral dsRNA was first described in *G. abietina* biotype A by Tuomivirta et al. [30]. They found two independent dsRNA banding patterns in a single *G. abietina* isolate, and later characterised one totivirus (Gremmeniella abietina RNA virus-lone 1, GaRV-L1) and two partitiviruses (Gremmeniella abietina RNA virus multisegmented, GaRV-MS1 and GaRV-MS2), but could not find dsRNA in ascospore isolates [31,32]. Narnaviruses were observed later, and are now known to be present in both biotypes of *G. abietina* [33,34]. Thereafter, biotype B has also been shown to host an endornavirus (Gremmeniella abietina RNA virus XL, GBRV-XL) [35] and, within the Spanish population, a taxonomically uncategorised mycovirus (Gremmeniella abietina RNA virus 6, GaRV6) [36,37].

### 2.1. Partitiviruses

Three full-length genomes have been characterised in three different strains of *G. abietina*, two in Finnish isolates of biotype A (GaRV-MS1 and 2) [31,32], and one in an isolate of the Spanish population of *G. abietina* (GaRV-MS1-3) [38]. GaMRV-MS1 has its genome divided into three segments, and the largest one contains the ORF (open reading frame) that codes for an RNA-dependent RNA polymerase (RdRp) and has a size of circa (ca.) 1.7 kb, the medium segment (ca. 1.5 kb) codes for a capsid protein (CP), and the smallest one (III) (ca. 1.1 kb) codes for a protein with unknown function (Figure 1A). The comparison of amino acid sequences of the CP, RdRp, and unknown protein revealed that the three full-length sequences described belong to the same species of the genus *Gammapartitivirus* [31,32,38]. Interestingly, GaRV-MS1 appears to have low genetic variability, and it is highly conserved not only in Europe, but also in North America [38]. GaRV-MS1 was detected in 28% of 162 investigated isolates. It primarily occurs in *G. abietina* biotype A (Table 1) but is also present in biotype B in Turkey [29]. When the occurrence of GaRV-MS1 was analysed within each population/biotype, the highest incidence was found in the Spanish population (56% of 50 isolates), followed by the biotype A population in North America (45% of 11 isolates), biotype A in most of Europe (16% of 68 isolates), and biotype B (and the Alpine biotype; in only 6% of 33 isolates). The virus GaMRV-MS1 evolves not only through purifying selection but also, to some extent, via recombination. Strain GaRV-MS1-2 seemed to be a recombinant between the complete CP sequences of GaRV-MS1-1 and GaRV-MS1-3, suggesting that GaRV-MS1-2 or one of its ancestors was a recombinant. Likewise, recombination was identified in the full-length RdRp sequences of GaRV-MS1-3 and 2, with GaRV-MS1-1 indicated to be a recombinant [38].

We reanalysed the taxonomic statuses of *G. abietina* viruses as the sequence information in GenBank is increasing quickly. This was done simply by using a BLAST search (amino acid sequences) to find the most similar *RdRp* genes, and MAFFT alignment using BLOSUM62 cost matrix to determine identities [39]. The reanalysis was carried out in October 2018. Based on this analysis, the closest relatives of GaRV-MS1 (and other partitiviruses of *G. abietina*) were 20 strains of Pseudogymnoascus destructans partitiviruses (PdPV-pa) [40] and Valsa malicola partitivirus (VmPV; (GenBank accession number AIS37554) with 77–78% identities. The closest strain of PdPV-pa (APG38267.1) with a partial sequence covering amino acid positions 53–362 of GaRV-MS1 RdRp had 78% identity to GaRV-MS1. The VmPV sequence was very short, and covered only amino acid positions 198–331 of GaRV-MS1 RdRP and had 76% identity with it.

### 2.2. Narnaviruses

Two separate narnavirus populations of the genus *Mitovirus* have been characterised and reported in *G. abietina*: Gremmeniella mitovirus 1 (GMV1) (originally named as GaMRV-S) and Gremmeniella mitovirus 2 (GMV2) (Table 1) [32,33,34]. They share 94% identity at aa-level, and have a typical mitovirus monopartite genome of around 2.5 kb and GC content of 30% (Figure 1B,C). Using the mitochondrial translation table, both of them code for a single large ORF of ca. 2 kb.

Based on the screening of the 2.5 kb band in 353 isolates of *G. abietina* [34], there was no evidence of mitoviruses in the six Swiss (Alpine biotype), two Turkish (biotype B), and six North American isolates (EU race or biotype A in North America). However, 68 of the 91 Spanish isolates harboured the dsRNA band. Similarly, among the 211 biotype A Finnish isolates analysed, 51 isolates (24%) carried a 2.5 kb dsRNA segment. Only three putative mitoviruses were found among the 37 Finnish biotype B isolates tested (8%).

The population genetic parameters calculated for the two populations suggest that GMV1 is genetically more variable than GMV2. The evolution of both GMV1 and GMV2 is mainly driven by mutation and selection, as no recombination events were detected [34]. Based on the comparison to sequences in the GenBank, GMV1 is most closely related to soybean leaf-associated mitovirus 4 (SlaMV4) [41], soybean leaf-associated mitovirus 2 (SlaMV2) [41], Alternaria arborescens mitovirus 1 (AaMV1) [42], and Alternaria brassicicola mitovirus 1 (AbMV1) [43]. The highest similarity was observed to the partial sequence of SSlaMV4, which was aligned with amino acid positions 1–529 of GMV1 RdRp with 44% identity. For the full sequences of SlaMV2, AaMV1, and AbMV1, RdRp had 43%, 42% and 42% identities with that of GMV1, respectively.

GMV1 is only observed in the Finnish biotype A and Spanish strains, whereas GMV2 infects the Finnish biotype B and the Spanish population. Thereby, the Spanish population of *G. abietina* harbours mitovirus strains that, in Finland, occur separately in biotype A and B strains. Therefore, the Spanish population is the first one hosting distantly related mycoviruses of a single genus in one population of *G. abietina*. This may suggest that horizontal transmission of viruses could have occurred between biotype B and the Spanish population (A type origin) in Spain, although biotype B has never been observed there. Furthermore, GMV2 has been observed only in one of the four studied localities in Spain, suggesting a certain local differentiation of virus populations among the Spanish *G. abietina* [34].

### 2.3. Totiviruses

GaRV-L1 and GaRV-L2 are putative members of the same species belonging to the genus *Victorivirus*. They were sequenced from two Finnish isolates belonging to biotype A [31]. Their genome lengths are ca. 5 kb, and show 90% overall identity (RdRp 98%). GaRV-L1 and 2 contain two large partially overlapping ORFs (Figure 1D). The first ORF starts at approximately nucleotide 270 from the 5’ end of the coding strand. Starting nucleotides for the second ORF were located in positions located 2.6 kb from the 5’ end in both cases. The protein encoded by the second ORF contained all eight conserved motifs of RdRps of viruses infecting lower eukaryotes [44]. Based on BLAST analysis using the RdRp amino acid sequence, the closest relatives of these viruses are Penicillium aurantiogriseum totivirus 1 (PaTV1) [45], Penicillium digitatum virus 1 (PdV1) [46], and Aspergillus mycovirus 178 (AMV178) [47]. PaTV1 and PdV1 covered the full RdRp sequence of GaRV-L1 with identities of 60% and 59%, whereas AMV178 covered positions 53–825 of GaRV-L1 RdRp sequence with an identity of 60%. The first ORF in both isolates coded for a putative CP, as BLAST searches indicated high similarity with analogous proteins of the viruses described above. 

Although there are no population studies available on GaRV-L, it should be noted that these viruses have only been observed in biotype A [31].

### 2.4. Endornaviruses

The virus with the largest genome found to date is GaBRV-XL, described in two Finnish isolates of *G. abietina* B type (Table 1) [35]. It is a linear, monopartite dsRNA virus of ca. 11 kb (Figure 1E), and belongs to the genus *Betaendornavirus*. The GaBRV-XL genomic structure has five conserved sites, and encodes for a putative 3249 aa polyprotein with four regions showing high similarity to putative viral methyltransferases, DExH box helicases, RNA helicase 1 of viruses, and RNA-dependent RNA polymerases. The closest relatives of GaBRV-XL, according to the BLAST search with RdRp motif of the polyprotein, are Discula destructiva virus 3 (DdV3) [48] and several endornaviruses from *Sclerotinia sclerotiorum* (including SsEV2-A) [49]. The DdV3 sequence covered only amino acid positions 107–165 in GaBRV-XL sequence with 72% identity, whereas SsEV2-A sequence covered the complete RdRp motif with 69% identity. In order to follow the International Committee on Taxonomy of Viruses (ICTV) rules, we propose renaming this virus as Gremmeniella betaendornavirus 1 (GBEV1).

### 2.5. Gremmeniella abietina RNA Virus 6

GaRV6 consists of a polymerase segment of ca. 2.1 kb with 54.7% GC content (Figure 1F) [36]. It seems to be part of an unclassified mycovirus group that is likely to be proposed as a novel virus family [37]. Its members are hosted by important plant pathogens, such as *Fusarium graminearum* [50], *Rhizoctonia solani* [51], *C. parasitica* (GenBank accession numbers KC549809 and KC549810), *H. annosum* [20], and the endophyte *Curvularia protuberata* [52]. The closest relatives of GaRV6 based on BLAST search using *RdRp* gene are Fusarium graminearum dsRNA mycovirus 5 (FgV5) [53] and Fusarium graminearum dsRNA mycovirus-4 (FgV4) [50]. The FgV-5 and FgV4 sequences covered amino acid positions 95–638 and 60–638 of GaRV6 RdRp with identities of 48% and 46%, respectively. This virus group is closely related to the *Partitiviridae* family [20,37,50]. Most of these viruses have bipartite genomes with sizes resembling those of partitiviruses (1700 to 2400 bp). The larger segment encodes the RdRp, whereas the smaller segment appears to have one or two ORFs coding proteins with unknown functions. However, the smaller genome segment of Ustilaginoidea virens RNA virus 4 (UvRV4) [54] and Heterobasidion RNA virus 6 (HetRV6) [20] have not been detected.

GaRV6 primarily appeared in the Spanish population of *G. abietina*, where its genetic diversity was minimal, despite its relatively high abundance (46% of 50 isolates screened) [36]. GaRV6 has also been detected by RT-qPCR in three other isolates belonging to biotype A in Italy, Canada, and Finland [55].

### 2.6. Multiple Virus Infections and Evidence of the Existence of Novel Viruses

*G. abietina* commonly hosts more than one virus in a single isolate [30,31,32,36]. In biotype A, up to five dsRNAs are found in some isolates [30], in B type three [35] and, in the Spanish population, there are up to eight different dsRNA-banding patterns described, some of which include up to eight bands (Figure 2), suggesting the existence of other *Gremmeniella* viruses not reported yet. Nothing is known about the interactions of multiple viruses co-inhabiting the same mycelia, nor of their effects on the host phenotype.

Here, we provide further information about the exceptional richness of the virus community hosted by Spanish isolates of *G. abietina* noted by Botella et al. [36]. Two dsRNA bands of putative viruses were detected in several isolates, including P3-7 and 06P (Figure 2) which were further analysed as previously (Appendix A). These isolates of *G. abietina* have been included in the previously mentioned studies about GaRV-MS1, GMV1 and GaRV6 [34,36,38].

#### 2.6.1. The ca. 6 kb dsRNA Band

A partial 2309 bp sequence of the ca. 6 kb dsRNA fragment (GenBank accession number LR031261; Figure 2) from isolate P3-7 was determined as previously described (Appendix A) [21,23,27,29]. The comparison in BLASTX (Table 2), resulted in the highest similarity (65%) with the RdRp of Macrophomina phaseolina single-stranded RNA virus 1 (MpSRV1) followed by Neofusicoccum luteum fusarivirus 1 (62%). These viruses are unclassified ssRNA viruses belonging to Fusariviridae, which is a recently proposed family not yet officially accepted by the ICTV. The sequence included a 332 bp conserved domain with an E-value of 3.34 × 10^−3^, belonging to a hypothetical ATP-dependent RNA helicase HrpB of *Haemophilus influenzae* (Accession number PRK11664).

An ORF with a length of 987 bp, coding for a protein of 328 aa, was detected. This sequence was obtained from a contig of 3 overlapping clones (682, 743 and 438 bps) and 5 reverse transcription (RT) PCR products. Specific primers (Appendix A) were designed based on the first contig achieved by cloning the 6 kb dsRNA band (Appendix A). The aa sequence had 61% identity with the MpSRV1 RdRp gene. We propose the name Gremmeniella fusarivirus 1 (GFV1) for this mycovirus.

A similar banding pattern has also been found in other isolates of *G. abietina* [56], such as 003P (Figure 2). However, the frequency of GFV1 in Spanish or other *G. abietina* populations has not been determined, with more specific methods such as reverse transcription (RT) PCR.

#### 2.6.2. The ca. 5 kb dsRNA Band

The 1852 bp and 1261 bp sequences (accession numbers LR030278 and LR030279) were determined from a 43-clone contig from a ca. 5 kb dsRNA fragment of Spanish isolate 06P (Figure 2). The BLASTX analyses with the two nucleotide sequences showed that they code for RdRp and CP proteins of a putative new virus of the genus *Victorivirus*. The BLASTX comparison of the partial RdRp sequence had the highest similarity (78%) with Sclerotinia nivalis victorivirus 1, followed by Aspergillus foetidus slow virus 1 (52%) (Table 3), but was distantly related to the previously known victoriviruses, GaRV-MS1 and GaRV-MS2 (39% and 38% identities in Blastx, respectively), from *G. abietina* biotype A. The partial RdRp fragment contains an ORF of 1557 bp that codes for a 512 aa protein.

In the case of the partial CP fragment, the consensus sequence derived from a seven-clone contig of 1261 bp also showed the highest similarity (81%) with Sclerotinia nivalis victorivirus 1, followed by Tolypocladium cylindrosporum virus 1 (68%) (Table 4). According to the 60% maximum identity criterion of the ICTV 9th report (2011) for the genus *Victorivirus*, this virus should be considered a strain of Sclerotinia nivalis victorivirus 1 because it shows 78% RdRp and 81% CP similarity to Sclerotinia nivalis victorivirus 1. Therefore, we designate the virus as Sclerotinia nivalis victorivirus 1—strain 1 from *Gremmeniella abietina* (SnVV1-Ga1)

A similar dsRNA banding pattern has also been found in other isolates of Spanish *G. abietina* [56], such as 003P and P3-7 (Figure 2), but its presence among *G. abietina* isolates has not been determined with specific methods.

## 3. Mycoviruses and Their Clarifying Message on the Complex Diversity of *G. abietina*

One of the fundamental questions in evolutionary biology is the degree to which the diversification of parasites and/or symbionts is associated with the diversification of their hosts [57]. Mycoviruses with RNA genomes produce mostly cryptic infections, replicate only in the fungal cytoplasm, and are transferred only through anastomosis and/or via sporal dispersal [58]. Such mechanisms would indicate that evolutionary patterns of mycoviruses can be expected to follow those of their hosts [59,60]. The diversification of the European race of *G. abietina* into different biotypes, and possible species [9], is mostly the outcome of geographical separation and ecological adaptation. Furthermore, its virus communities provide insights about this history of dispersion, with a special relevancy to the establishment of new populations in Spain and Turkey.

The European race of *G. abietina* is known to host only one species of gammapartitivirus (GaRV-MS1) [38]. In northern Europe, it has only been found in biotype A isolates, but it occurs also in Turkish isolates, which belong to biotype B (Table 1) [16,34]. On the other hand, the lack of GaRV-MS1 in biotype B in northern Europe and its occurrence in Turkey suggests that the virus transmission must have occurred between biotypes A and B during their evolutionary history. This virus transmission may have happened prior to the introduction of *G. abietina* to Turkey or, alternatively, in Turkey with an unrecognised biotype A population. In the first scenario, biotype B isolates arrived in Turkey with GaRV-MS1 viruses and, in the second scenario, they would have received the viruses in Turkey. A third possibility is that GaRV-MS1 viruses occurred in both biotypes during their evolutionary history, but were later lost in B-type with the exception of the Turkish population. The presence of GaRV-MS1 in the European race (biotype A) in North America, introduced from Europe in 1977 [61], indicates that the virus travelled with its host, and has not changed much over these years.

The virus community inhabiting the Spanish *G. abietina* population supports its origin as biotype A [16] because (i) recombination events have been detected between three strains of GaRV-MS1 present in Finland and Spain and (ii) GaRV6 primarily appears in Spain, but was also found in single isolates of biotype A from Canada, Italy and Finland, and (iii) the Spanish population also harbours both mitovirus species GMV1 and GMV2, associated with biotype A and B strains in northern Europe, respectively.

The occurrence of GMV2 in both biotype B and the Spanish population, together with its extremely low genetic variability, would be in accordance with a recent host switch and subsequent adaptation to a new host [34]. Natural interspecies transmission of viruses has been shown for *Cryphonectria* sp. and *Cryphonectria parasitica* [22], *Sclerotinia homoeocarpa* and *Ophiostoma ulmi* [23], *O. ulmi* and *Ophiostoma novo-ulmi* [24], and within species of the genera *Heterobasidion* [20,25,26], *Sclerotinia* [27,28], and *Rhizoctonia* [29]. However, biotype B of *G. abietina* has never been shown to exist in Spain and, therefore, the timing and place of this possible host jump remains open. It could have occurred in the past, or even today, in the mountains of northern Spain, where the snow cover in wintertime is enough to fulfil its need [12,13,62]. Further light on this issue would be obtained by a more detailed study on the presence of biotype B in possible candidate areas.

No evidence is available for the presence of mitoviruses in the Swiss (Alpine biotype), Turkish (biotype B), and North American isolates (biotype A). Therefore, the Spanish population of *G. abietina* is the only one hosting the two different mitoviruses, which underlines its unique position within the taxonomy of *G. abietina* var. *abietina*, extending also to its mycovirus community.

A certain level of specialisation is detected in Finland [31,32,35], where both *G. abietina* biotypes (or species) cohabit. In Finland, viruses occurring in biotype A and biotype B diverge (Table 1). The Finnish biotype A is known to host four viruses (GaRV-MS1, GMV1, GaRV6, and GaRV-L) while, within the *G. abietina* biotype B, two different viruses (GMV2 and GBEV1) have been reported [35].

Interestingly, we report, here, that the Spanish population hosts a fusarivirus, GFV1, and also an unrelated victorivirus, SnVV1, the presence of which, in other populations of *G. abietina*, has not been tested directly, but the lack of corresponding dsRNA banding patterns in Finnish populations suggests its unlikeliness. Therefore, it can also be hypothesised that these viruses represent transmission from some distantly related fungi, such as *Sclerotinia nivalis*, to the Spanish population of *G. abietina*. This possibility is supported by the emerging evidence of virus transmissions over wide taxonomic distances known from other fungi [60,63,64,65] and, therefore, should be tested in the future.

Altogether, phylogenetic and prevalence studies of the distinct viruses provide evidence that support separation of biotypes A and B from each other, as well as on the origin of separate populations of *G. abietina* in North America, Spain, and Turkey.

## 4. Consequences of Viral Infections on *G. abietina*

Two pathogenicity tests with *G. abietina* biotype A isolates harbouring GaRV-L and GaRV-MS dsRNA patterns were performed, but no firm conclusion could be drawn on the effect of dsRNA viruses on *G. abietina* pathogenicity towards *Pinus sylvestris* [32]. All isolates were pathogenic to host plants, but both dsRNA-containing and dsRNA-free isolates were found among the most pathogenic isolates, and there was no difference between the pathogenicity of isolates containing GaRV-L and GaRV-MS patterns. However, it should be noted that no pairs of isogenic dsRNA-containing and dsRNA-free isolates were available for the pathogenicity tests. Additionally, the inoculated trees were genetically different, and the pathogenicity test setup could only measure how mycelium is able to grow in phloem after inoculation [32].

Romeralo et al. [66] noted that the existence of GMV1 appeared to have a substantial increasing effect on the growth rate of *G. abietina* at its optimum growing temperature, 15 °C. Whether this increase in mycelial growth is connected to the virulence of their hosts has not been tested. In contrast, GaRV6 occurrence had a negative impact on the growth rate of *G. abietina* on artificial media [45].

Based on the results described above, at least some viruses may have effects on the phenotype of *G. abietina*. However, conclusions about the real and specific role of each type of these viruses cannot be fully determined, since multiple virus infections are common in this conifer pathogen [30,31,32,35,36,55,56]. For example, isolate P3-7, has a large viral dsRNA load (Figure 2). It hosts GaRV6, GaRV-MS1, and GMV1, as well as both novel species of *Fusarivirus* and *Victorivirus* reported in this review, and had the lowest growth rate in the experiment by Botella et al. [55]. Conversely, in the same study, isolate 06P had the highest growth rate in the three temperatures tested, and it only hosts the possible victorivirus reported here. Thus, it is very necessary to carry out experiments, including pairs of cured and single virus infected isogenic *G. abietina* isolates; unfortunately, up to now, virus removal trials with higher temperatures or cycloheximide have failed with this pathogen [55]. Furthermore, it has not been possible to infect viruses of *G. abietina* in dual culture experiments used for *C. parasitica* since the 1960s (and, thereafter, for many other fungi) and, therefore, artificial transmission methods should also be developed for *G. abietina* viruses.

## 5. Conclusions

*G. abietina* hosts a distinct and abundant community of viruses belonging to different genera and families of mycoviruses. Its diversification and spread in different countries has also affected the divergence of its virus community. Based on a single study, it appears that multiple virus infections may affect the phenotype of the host, but the effects of single viruses remain obscure, and should be determined by in vivo and in vitro experiments using isogenic pairs of single-infected and virus-free isolates.

## Figures and Tables

**Figure 1 viruses-10-00654-f001:**
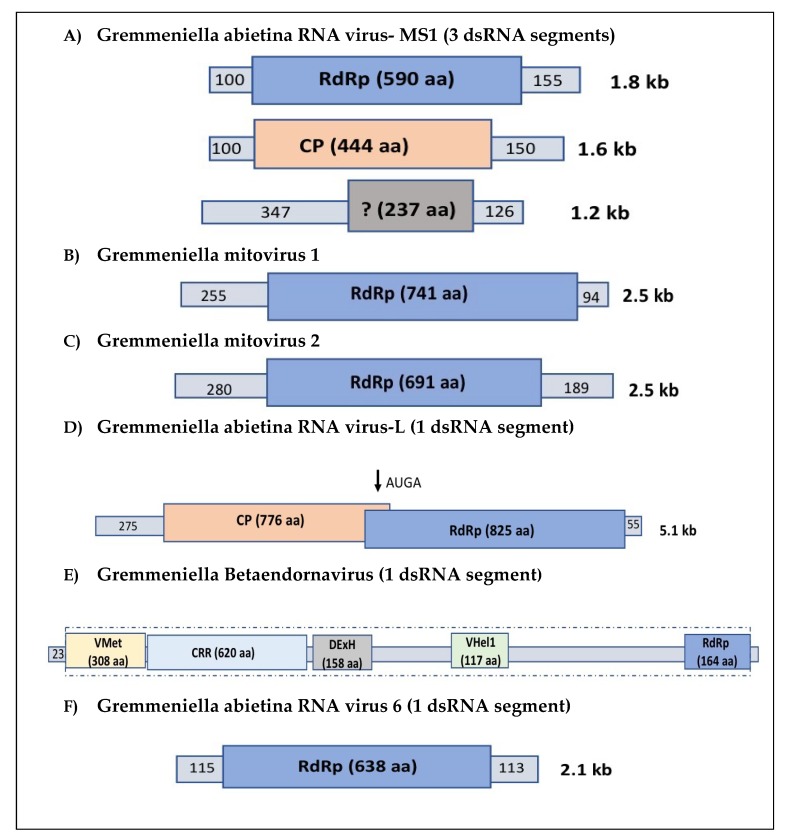
Schematic illustration of the genomic organisation of all fully sequenced *G. abietina* mycoviruses. ORFs and conserved domains are represented by rectangular coloured boxes indicating the coded proteins and the size of their amino acid sequences. They are flanked by 5′- and 3′-UTRs, numbers represent base pairs (bp). The full-length dsRNA segment size is indicated on the right. (**A**) Gremmeniella abietina RNA virus-MS1; (**B**) Gremmeniella mitovirus 1 (previously named Gremmeniella abietina RNA virus-S1); (**C**) Gremmeniella mitovirus 2; (**D**) Gremmeniella abietina RNA virus-L; (**E**) Gremmeniella betaendornavirus 1 (previously Gremmeniella abietina RNA virus-XL) consists of a polyprotein of 3249 and five conserved regions: viral methyltransferase (VMet), cysteine-rich region (CRR), DExH box helicase (DExH), viral RNA helicase 1 (VHel1), and RNA_dep_RNApol2 (RdRp); (**F**) Gremmeniella abietina RNA virus 6.

**Figure 2 viruses-10-00654-f002:**
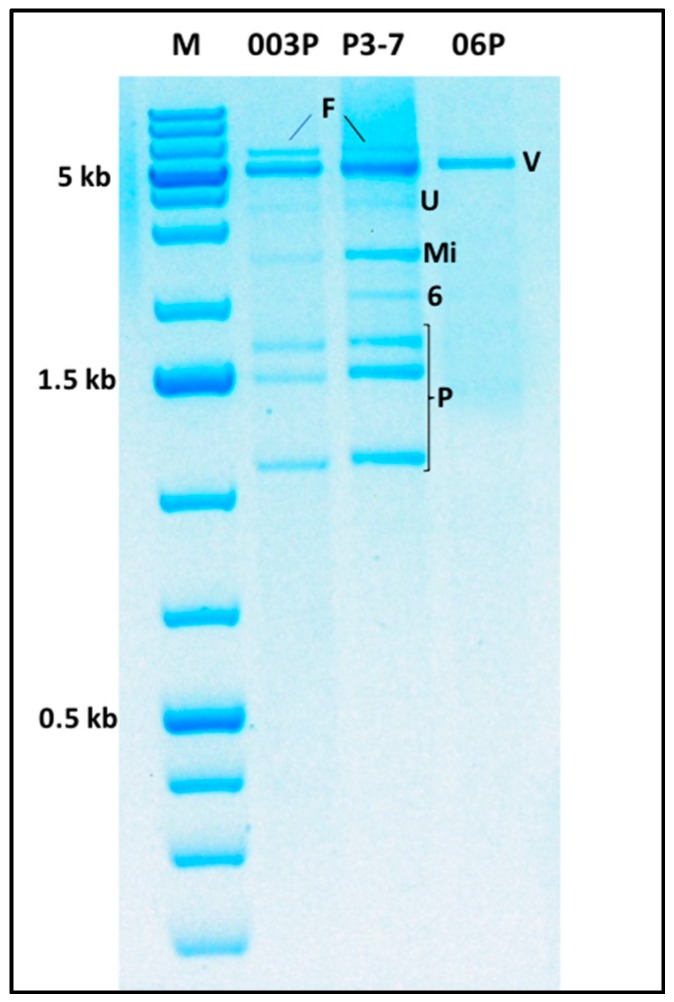
dsRNA banding pattern of the isolates 003P, P3-7, and 06P of Gremmeniella abietina after DNase I and S1 nuclease treatment. M, DNA marker (GeneRuler 1 kb Plus DNA Ladder, 75–20,000 bp, Thermo Scientific); **F**, dsRNAs corresponding to a putative fusarivirus; **V**, dsRNA corresponding to a putative victorivirus, **U**, unknown band, **Mi**, dsRNA corresponding to Gremmeniella mitovirus 1; **6**, dsRNA band corresponding to GaRV6; **P**, dsRNA corresponding to Gremmeniella abietina RNA virus MS1.

**Table 1 viruses-10-00654-t001:** Described viruses in *G. abietina*.

Virus Name	Virus Abbreviation	Full-Length Virus Strains	GenBank Accessions *	Virus Genus	Genome	Fungal Population	Population Studies	Countries Detected	References
**Gremmeniella abietina** RNA virus multi segmented 1	GaRV-MS1	3	KJ786411-KJ786413, AY089993-5, AY615211-13,	*Gammapartitivirus*	dsRNA	A, B, SP	YES	Canada, USA, Finland, Spain, Montenegro, Italy, Turkey	[31,32,38]
**Gremmeniella abietina** mitovirus 1	GMV1	3	HE586988, AF534641, AY615209	*Mitovirus*	(+) ssRNA	A, SP	YES	Finland, Spain	[32,33,34]
**Gremmeniella abietina** mitovirus 2	GMV2	1	JN654496	*Mitovirus*	(+) ssRNA	B, SP	YES	Finland, Spain	[34]
**Gremmeniella abietina** RNA virus lone	GaRV-L	2	AF337175, AY615210	*Totivirus*	dsRNA	A	NO	Finland	[31,32]
**Gremmeniella abietina** RNA virus XL	GaBRV-XL	2	DQ399289-90	*Betaendornavirus*	dsRNA	B	NO	Finland	[35]
**Gremmeniella abietina** RNA virus 6	GaRV6	1	KJ742567.1	Unassigned ^†^	dsRNA	A, SP	YES	Finland, Canada, Italy, Spain	[36]

^†^ Virus belonging to a new family not defined and submitted in the ICTV yet. * GenBank accession numbers of full-length virus sequences.

**Table 2 viruses-10-00654-t002:** BLASTX comparison of a 2228 bp RdRp sequence obtained from cloning 6 kb dsRNA of P3-7 *G. abietina* isolate.

GenBank BLASTX Database	Sequence ID	Family	Identities	Matching Region ^1^
Macrophomina phaseolina single-stranded RNA virus 1	ALD89094.1	*Fusariviridae **	217/335 (65%)	512 to 845
Neofusicoccum luteum fusarivirus 1	ARO52688.1	*Fusariviridae **	205/333 (62%)	527 to 856
Penicillium aurantiogriseum fusarivirus 1	YP_009182154.1	*Fusariviridae **	198/328 (60%)	511 to 838
Penicillium roqueforti ssRNA mycovirus 1	YP_009052456.1	*Fusariviridae **	177/313 (57%)	500 to 812
Fusarium graminearum dsRNA mycovirus-1	YP_223920.2	*Fusariviridae **	179/334 (54%)	530 to 863
Sodiomyces alkalinus fusarivirus 1	ATP75827.1	*Fusariviridae **	183/326 (56%)	445 to 770
Rosellinia necatrix fusarivirus 1	YP_009047147.1	*Fusariviridae **	80/329 (55%)	522 to 850
Pleospora typhicola fusarivirus 1	YP_009182158.1	*Fusariviridae **	178/329 (54%)	521 to 849
Fusarium poae fusarivirus 1	YP_009272906.1	*Fusariviridae **	172/326 (53%)	496 to 818
Agaricus bisporus virus 11	AQM49938.1	*Fusariviridae **	151/314 (48%)	592 to 901

* Proposed family not official in the ICTV yet; ^1^ Matching region in the aa sequence of the database sequences.

**Table 3 viruses-10-00654-t003:** BLASTX comparison of a 1852 bp RdRp sequence obtained from cloning 5 kb dsRNA of 06P *G. abietina* isolate.

GenBank BLASTX database	Sequence ID	Family	Identities	Matching Region ^1^
Sclerotinia nivalis victorivirus 1	YP_009259368.1	*Totiviridae*	456/587 (78%)	247 to 833
Aspergillus foetidus slow virus 1	YP_009508249.1	*Totiviridae*	306/584 (52%)	254 to 837
Beauveria bassiana victorivirus 1	AMQ11131.1	*Totiviridae*	302/585 (52%)	258 to 841
Sphaeropsis sapinea RNA virus	NP_047558.1	*Totiviridae*	314/587 (53%)	254 to 837
Rosellinia necatrix victorivirus 1	BAM36400.1	*Totiviridae*	397/587 (67%	107 to 692
Beauveria bassiana victorivirus NZL/1980	YP_009032633.1	*Totiviridae*	301/585 (51%)	258 to 841
Soybean-associated double-stranded RNA virus 1	ALM62239.1	*Totiviridae*	311/587 (53%)	255 to 840
Ustilaginoidea virens RNA virus 1	AGO04407.1	*Totiviridae*	302/584 (52%)	243 to 825
Bipolaris maydis victorivirus 1	AXB26764.1	*Totiviridae*	289/588 (49%)	253 to 835
Botryotinia fuckeliana totivirus 1	YP_001109580.1	*Totiviridae*	291/581 (50%)	266 to 836

^1^ Matching region in the aa sequence of the database sequences.

**Table 4 viruses-10-00654-t004:** BLASTX comparison of a 1261 nt capsid protein (CP) sequence obtained from cloning 5 kb dsRNA of 06P *G. abietina* isolate.

GenBank BLASTX Database	Sequence ID	Family	Identities	Matching Region ^1^
Sclerotinia nivalis victorivirus 1	YP_009259367.1	*Totiviridae*	293/363 (81%)	1 to 363
Tolypocladium cylindrosporum virus 1	YP_004089629.1	*Totiviridae*	247/362 (68%)	1 to 362
Beauveria bassiana victorivirus NZL/1980	YP_009032632.1	*Totiviridae*	237/361 (66%)	1 to 361
Helminthosporium victoriae virus 190S	NP_619669.2	*Totiviridae*	240/363 (66%)	1 to 363
Bipolaris maydis victorivirus 1	AXB26763.1	*Totiviridae*	240/363 (66%)	1 to 363
Botryotinia fuckeliana totivirus 1	YP_001109579.1	*Totiviridae*	227/363 (63%)	1 to 363
Aspergillus foetidus slow virus 1	YP_009508248.1	*Totiviridae*	221/360 (61%)	1 to 360
Fusarium asiaticum victorivirus 1	AYD49681.1	*Totiviridae*	225/364 (62%)	1 to 363
Ustilaginoidea virens RNA virus L	YP_009094184.1	*Totiviridae*	223/363 (61%)	1 to 363
Rosellinia necatrix victorivirus 1	BAM36399.1	*Totiviridae*	227/362 (63%)	1 to 361
Sclerotinia sclerotiorum victorivirus 1 (partial)	AWY10937.1	*Totiviridae*	208/263 (79%)	1 to 233

^1^ Matching region in the aa sequence of the database sequences.

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
