# Peer review of "Description, Distribution, and Relevance of Viruses of the Forest Pathogen Gremmeniella abietina"

_viruses, 2018, doi:10.3390/v10110654_

Round 1

Reviewer 1 Report

General comment

This paper is generally well presented and provides a useful review of the viruses currently known to infect the fungus Gremmeniella abietina.  However, the discussion and conclusions, relating to the viruses “relevance in claryifying the taxonomy of G. abietina”, appears to be mostly speculative. In particular, although for some viruses the authors give % incidence, the authors do not state the sample size or the number of different sampling sites.  This is particularly relevant for those viruses that are assumed to be absent, where the lack of detection could be the result of small sample numbers or limited sampling sites. I recommend that the discussion and conclusions be revised to give a better indication of the number and range of samples on which the various observations and conclusions are based.

Specific comments

The full names of the viruses are not provided in the main text, only acronyms are used. At first mention the full name (plus acronym) should be used.

L64-69 – what are the sample sizes from which the % incidences are calculated?

L69-71 – it is not obvious how the preceding information (L55-68) leads to the conclusion that GaRV-MS1 “does not have a direct impact on the genetic structure of the host”. This needs to be explained.

L88-89 – Isn’t the evolution of everything “mainly driven by mutation and selection”? What particular point are the authors trying to make here?

L95 – I suggest changing the wording to “…… family not defined and submitted….”

L103-104 – for the statement that “GMV2 has been observed only in a single locality in Spain” to have any relevance it requires information on the number of localities sampled and the number of samples tested.

L123 – what is meant by “seems to have five conserved sites”? 

L171 – 46% of what sample size?

L183-184  – what was the outcome of the previous investigation for the presence of these viruses? It’s not very informative simply to say they had been investigated.

L212 – need to add accession numbers

L223  – proposing GVV3 as a new species is not in line with ICTV criteria for victorivirus species. The ICTV 9th report (2011) states “the amino acid sequence identity in pairwise comparisons of either the CP or the RdRp gene product of the nine species is no more than 60%”. Since GVV3 shows 78% RdRp and 81% CP similarity to Sclerotinia nivalis victorivirus 1, based on the above criteria it should be considered a strain of Sclerotinia nivalis victorivirus 1.

Section 3 (L234-288) – the fungal names in this section should be in italics (currently none of them are).

L248-253 – this section proposes several possible scenarios for the presence of GaRV-MS1 in G. abientinabiotype B in Turkey but provides no evidence to support any of these possibilities, consequently it adds little to the discussion.

L273 – on what sample size is the “no evidence” based?

L279  - the text mentions “three” species but four are listed.

L284 – what does “relatively well studied” mean? Can you quantify this?

L320 – it appears that the statement “it looks like high virus loads might be affecting the phenotype of 
the host” is based on the single study of Botella et al. (2017). Consequently its validity as a general conclusion is highly questionable.

Author Response

General comment

This paper is generally well presented and provides a useful review of the viruses currently known to infect the fungus Gremmeniella abietina.  However, the discussion and conclusions, relating to the viruses “relevance in claryifying the taxonomy of G. abietina”, appears to be mostly speculative. In particular, although for some viruses the authors give % incidence, the authors do not state the sample size or the number of different sampling sites.  This is particularly relevant for those viruses that are assumed to be absent, where the lack of detection could be the result of small sample numbers or limited sampling sites. I recommend that the discussion and conclusions be revised to give a better indication of the number and range of samples on which the various observations and conclusions are based.

Specific comments

The full names of the viruses are not provided in the main text, only acronyms are used. At first mention the full name (plus acronym) should be used. The full names of the viruses were provided as suggested.

L64-69 – what are the sample sizes from which the % incidences are calculated? As indicated, we included the number of isolates analysed for that study.

L69-71 – it is not obvious how the preceding information (L55-68) leads to the conclusion that GaRV-MS1 “does not have a direct impact on the genetic structure of the host”. This needs to be explained. We preferred to delete this sentence here because it is not connected to the rest of the paragraph.

L88-89 – Isn’t the evolution of everything “mainly driven by mutation and selection”? What particular point are the authors trying to make here? We mean that no recombination events were detected, as it happened in the study about the gammapartitivirus.

L95 – I suggest changing the wording to “…… family not defined and submitted….” We changed the text as suggested

L103-104 – for the statement that “GMV2 has been observed only in a single locality in Spain” to have any relevance it requires information on the number of localities sampled and the number of samples tested. We have included the information required.

L123 – what is meant by “seems to have five conserved sites”? We corrected this sentence

L171 – 46% of what sample size? As indicated, we included the number of isolates analysed for all the studies.

L183-184 – what was the outcome of the previous investigation for the presence of these viruses? It’s not very informative simply to say they had been investigated. We tried to be clearer in this point

L212 – need to add accession numbers. The accession numbers were added to the text.

L223  – proposing GVV3 as a new species is not in line with ICTV criteria for victorivirus species. The ICTV 9th report (2011) states “the amino acid sequence identity in pairwise comparisons of either the CP or the RdRp gene product of the nine species is no more than 60%”. Since GVV3 shows 78% RdRp and 81% CP similarity to Sclerotinia nivalis victorivirus 1, based on the above criteria it should be considered a strain of Sclerotinia nivalis victorivirus 1. We appreciate this comment very much and we have included it in the text.

Section 3 (L234-288) – the fungal names in this section should be in italics (currently none of them are). We apologise for these errors, they happened when integrating the text into the manuscript template of Viruses.

L248-253 – this section proposes several possible scenarios for the presence of GaRV-MS1 in G. abietina biotype B in Turkey but provides no evidence to support any of these possibilities, consequently it adds little to the discussion. If possible, we would like to keep the discussion of these scenarios because although speculative they can be the hypotheses of further future studies.

L273 – on what sample size is the “no evidence” based? We included this information in the section about narnaviruses

L279- the text mentions “three” species but four are listed. We apologize for this mistake and we amended it.

L284 – what does “relatively well studied” mean? Can you quantify this? We modified this sentence.

L320 – it appears that the statement “it looks like high virus loads might be affecting the phenotype of the host” is based on the single study of Botella et al. (2017). Consequently, its validity as a general conclusion is highly questionable. We slightly modified this sentence but we consider that our conclusions must be as open as they are in the text because we do not have further results to be more explicit.

Reviewer 2 Report

Comments and Suggestions for Authors

The paper by Leticia Botella and Jarkko Hantula provides a concise update on viruses associated with Gremmeniella abietina, a forest pathogen causative agent of shoot blight and stem canker. It will be of interest to the field of mycovirology.

Below are listed major and minor comments.

Please give an overview of a number of the tested fungal isolates from the cited publications. 

Although using BLAST for comparative analyses is convenient, it is difficult to estimate true differences before compared sequences as the coverage of aligned regions was not provided. It would be better to re-align the sequences using CLUSTAL, MAFFT or any other alignment tool.

Clarifying message of mycoviruses on the complex diversity of G. abietina is not clear. Please add compact conclusion at the end of the section.

All virus nomenclature should follow ICTV guidelines, see comments for the lines 94 - 96. Similarly, all latin names should be italicized.

The paper needs careful English editing and proofreading. 

Minor comments:

line 79: please indicate a date of performed analyses

lines 94 - 96: check virus names and abbreviations here and elsewhere. For example, "Gremmeniella abietina RNA virus MS1" should be used instead of "Gremmeniella abietina RNA virus multi segmented 1".

Gamm"a"partitivirus

lines 101 - 103: information about horizontal as well as interspecies transmission, mentioned at lines 286 - 287, should be given in the introduction section (somewhere after lines 42 - 43)

line 121: rather "with the biggest genome"

line 126: "viruses BLAST" - something is missing

line 127: "polygene" is not correctly used, maybe "polyprotein"?

lines 144 - 160: although it was stated that the figure is schematic, some domains had different sizes (for the endornavirus). Either use same size for all boxes or prepare scaled figure. Use same font sizes. Do not partially abbreviate virus names, like "G. abietina RNA virus-S".

lines 174 - 175: correct the "Botella et al. 2012b" citation

line 182: "ds"RNA bands

line 183: "have" previously been

lines 189 - 191: remove italic formatting

line 193: accession number is missing.

line 196: remove "the" before Macrophomina

line 199: E-value is not similarity

line 202, 213, 221    give a description what is a 3-clone contig

line 202: reverse transcript"ion"

line 212: accession "n"umbers. The numbers are missing.

line 215: the genus Victorivirus

lines 237 - 238: what about mitoviruses?

line 240: see comment for lines 101 - 103

line 278: "There, viruses occurring in biotype A and biotype B diverge" – something is missing

line 279: four, not three, viruses are listed. Delete "species".

references 25 and 31 are identical

line 320: maybe multiple infections should be used instead of "high virus loads"

Author Response

Reviewer 2

Comments and Suggestions for Authors

The paper by Leticia Botella and Jarkko Hantula provides a concise update on viruses associated with Gremmeniella abietina, a forest pathogen causative agent of shoot blight and stem canker. It will be of interest to the field of mycovirology.

Below are listed major and minor comments.

-       Please give an overview of a number of the tested fungal isolates from the cited publications. As indicated, we have included the number of isolates analysed for all the studies

-       Although using BLAST for comparative analyses is convenient, it is difficult to estimate true differences before compared sequences as the coverage of aligned regions was not provided. It would be better to re-align the sequences using CLUSTAL, MAFFT or any other alignment tool. As indicated, we have realigned all the sequences using MAFFT.

-       Clarifying message of mycoviruses on the complex diversity of G. abietina is not clear. Please add compact conclusion at the end of the section. We included a final paragraph in this section in order to be clearer in this point.

-       All virus nomenclature should follow ICTV guidelines, see comments for the lines 94 - 96. Similarly, all latin names should be italicized. We revised all the names and italicized all the latin names that were not.

-       The paper needs careful English editing and proofreading. The manuscript was edited and corrected by an English native speaker prior to its resubmission.

Minor comments:

line 79:           please indicate a date of performed analyses. The comparisons were carried out in September 2018.

lines 94 - 96:  check virus names and abbreviations here and elsewhere. For example, "Gremmeniella abietina RNA virus MS1" should be used instead of "Gremmeniella abietina RNA virus multi segmented 1". We changed the name in the table as suggested in this point.

Gamm"a"partitivirues. This mistake was amended

lines 101 - 103:          information about horizontal as well as interspecies transmission, mentioned at lines 286 - 287, should be given in the introduction section (somewhere after lines 42 - 43). We added a paragraph about interspecies transmission in the introduction.

line 121:         rather "with the biggest genome". Corrected as suggested

line 126:         "viruses BLAST" - something is missing. We corrected this error

line 127:         "polygene" is not correctly used, maybe "polyprotein"? We corrected it as suggested

lines 144 - 160:          although it was stated that the figure is schematic, some domains had different sizes (for the endornavirus). Either use same size for all boxes or prepare scaled figure. Use same font sizes. Do not partially abbreviate virus names, like "G. abietina RNA virus-S". Figure 1 was improved following the editor’s comments.

lines 174 - 175:          correct the "Botella et al. 2012b" citation. We apologize for this mistake and we amended it.

line 182:         "ds"RNA bands. We corrected it as suggested

line 183:         "have" previously been. We apologize for this mistake and we amended it.

lines 189 - 191:          remove italic formatting We apologize for this mistake and we amended it.

-       line 193:   accession number is missing. The accession numbers were added to the text.

line 196:         remove "the" before Macrophomina. We corrected it as suggested

line 199:         E-value is not similarity. We corrected the text

line 202, 213, 221    give a description what is a 3-clone contig. More detailed information was given

line 202:         reverse transcript"ion" This mistake was amended

line 212:         accession "n"umbers. The numbers are missing. The accession numbers were added to the text.

line 215:         the genus Victorivirus. We included this sentence as suggested

lines 237 - 238:          what about mitoviruses? Mitoviruses are spread intracellularly. No evidences of extracellular stages have been described so far.

line 240:         see comment for lines 101 - 103

line 278:         "There, viruses occurring in biotype A and biotype B diverge" – something is missing. We rewrote this sentence in order to be clearer

line 279:         four, not three, viruses are listed. Delete "species". Corrected as suggested

references 25 and 31 are identical. Citations and references have been updated

line 320:         maybe multiple infections should be used instead of "high virus loads" We changed this senten